# A Rheological Analysis of Biomaterial Behaviour as a Tool to Detect the Dilution of Heather Honey

**DOI:** 10.3390/ma14102472

**Published:** 2021-05-11

**Authors:** Antonín Přidal, Petr Trávníček, Jan Kudělka, Šárka Nedomová, Sylvie Ondrušíková, Daniel Trost, Vojtěch Kumbár

**Affiliations:** 1Department of Zoology, Fisheries, Hydrobiology and Apidology, Faculty of AgriSciences, Mendel University in Brno, Zemedelska 1, 613 00 Brno, Czech Republic; apridal@mendelu.cz; 2Department of Agricultural, Food and Environmental Engineering, Faculty of AgriSciences, Mendel University in Brno, Zemedelska 1, 613 00 Brno, Czech Republic; petr.travnicek@mendelu.cz (P.T.); jan.kudelka@mendelu.cz (J.K.); 3Department of Food Technology, Faculty of AgriSciences, Mendel University in Brno, Zemedelska 1, 613 00 Brno, Czech Republic; sarka.nedomova@mendelu.cz (Š.N.); sylvie.ondrusikova@mendelu.cz (S.O.); 4Department of Technology and Automobile Transport (Section Physics), Faculty of AgriSciences, Mendel University in Brno, Zemedelska 1, 613 00 Brno, Czech Republic; xtrost@mendelu.cz

**Keywords:** heather honey, dilution, rheology, time-dependent behaviour, comparison rheological parameter, mathematical modelling

## Abstract

Heather honey is a valuable and rheologically special type of honey. Its above-average selling price may motivate its intentional violation with a mixture of honey from another botanical origin, the price of which is lower on the market. This work deals with the rheological properties of such devalued heather honey in order to determine the changes in the individual rheological parameters depending on the degree of dilution of the heather honey. For this purpose, a differently diluted heather honey sample series was created and the following rheological parameters were determined: hysteresis area, *n*-value, yield stress (*τ*_0_), parameter B (Weltman model), parameter *ϕ*, or parameter C (model describing the logarithmic dependence of the complex viscosity on the angular frequency). Part of the work was research into whether the set parameters can be used as comparative parameters. It was found that the hysteresis area does not appear to be a suitable relative comparison parameter due to the high variability. The parameters that appear to be suitable are the relative parameters *n*-value and the parameter *ϕ*, which showed the greatest stability. The change in the determined rheological parameters is, depending on the degree of dilution, non-linear with a step change between the samples containing 40% (*w*/*w*) and 60% (*w*/*w*) of a heather honey.

## 1. Introduction

Honey is a natural food product and is considered to be the oldest sweetener. It is a very good source of many valuable nutrients that are beneficial to the human body. Honey has antioxidant, antibacterial, antifungal and antiviral properties, which are also used in the so-called apitherapy [1]. The antibacterial activity of honey is caused by its low acidity, high osmolarity and the content hydrogen peroxide and other components, such as polyphenolic compounds [2]. Ling heather honey (hereinafter “heather honey”) is a special type of honey arising from nectar of *Calluna vulgaris*. This honey is highly valued for its characteristic strong taste, health properties and unusual texture characteristics. It is a low-growing shrub predominantly on acid soils in the sun or semi-shade and blooms in late summer [3]. This plant species has a high beekeeping value. Honeybees from its nectar produce honey with special sensory properties, such as an amber to dark colour, a plant fragrance reminiscent of a moist ground and a persistent taste of bitterness, which are highly appreciated by consumers [4].

The price of heather honey on the market is considerably higher than common honey for reasons of it’s the exceptionality and difficulty in harvesting. This also applies to other special honeys, such as manuka honey (*Leptospermum scoparium*), which is one of the highly priced honeys [5]. Moreover, climatic change is bound to alter the conditions in heathlands and increase the risk of a reduced honey production [6]. It can, therefore, be assumed that these special honeys can be the target of fraud. Artificial dilutions are used for the violation of common honeys [7]. Alternatively, a common honey can be used for dilution of the special honeys. There are standards where the parameters are stated which define special honeys [8]. Various analytical methods can be used to determine these parameters and characterise the honey [9]. The determination of the rheological properties can be considered one of the indirect methods [10,11]. The fact that some special honeys show a different rheological behaviour, compared to common honeys, is used as an indicator in this area.

It is well known that some honey types in dependence on the botanical origin were classified as non-Newtonian fluids with a thixotropy behaviour [12]. Heather honey is one of them [1,13]. There is a presumption that, in the case of diluting heather honey with common honey, there will be a change in the rheological properties. The rheological behaviour of a heather honey mixed with a common honey has not been studied. Determination of the rheological properties of such mixtures can be a support method for the detection of a honey’s adulteration to protect consumers.

There are many works dealing with the problem of honey adulteration [7,14,15,16,17] and less works, where the knowledge of rheology is used for this purpose [11,18,19,20,21,22,23].

This study describes rheological properties and their changes in the heather honey in dependence of its experimental dilution by a honey with the Newtonian properties.

## 2. Materials and Methods

### 2.1. Samples

The lime (*Tilia* spp.) and ling heather honey (*Calluna vulgaris*, hereinafter “heather honey”) were used. The lime honey was produced at apiary of the first author (Příbram na Moravě, Czech Republic) and the ling heather honey was processed in the company Honningcentralen (Kløfta, Norway). Initially, both samples of original honey were de-crystallised in a thermostat and heated up to 50 °C. The warming-up lasted until the crystals in the honey were melted. Samples of the above and below defined original honeys (Table 1) as well as the mixed samples were prepared for the rheological tests. The melted honeys were mixed in a weight concentration with an accuracy 0.01 mg. The mixed samples contained 80% (S80), 60% (S60), 40% (S40), 20% (S20) and 10% (*w/w*) (S10) of the heather honey. The sample identification for pure lime honey is S0 and for pure heather honey is it S100.

### 2.2. Characterisation of Honey

The water content in the samples was measured by a refractive index at a temperature of 20–22 °C with extrapolation to 20 °C (ABBE refractometer, Carl Zeiss, type G, Jena, Germany) [24]. The electrolytic conductivity was measured with a Mitronic MVM-1 device at room temperature of 22 °C with extrapolation to 20 °C according to the correction factor described by [24].

The botanical origins of the honey samples were evaluated by melissopalynological analysis, in accordance with the harmonised methodology developed by [25] and other supporting protocols by [8,26,27]. The pollen grains were identified with the pollen collection at Mendel University in Brno, and the pollen grain catalogue by [28]. One tetrad of heather pollen grains was counted as a one pollen grain. The sensory properties of the samples (colour, odour and taste) corresponded with results of the pollen analysis. The geographical origin of the honey samples was not necessarily verified due to the honey samples coming from known apiaries. The honeydew elements were evaluated subjectively as follows: (1) The abundance of clusters, and (2) the abundance of fragments of algae and fungi. Their abundance was determined on the following subjective scale: Absent, sporadic, few, moderate and widely abundant.

The water content and the electrolytic conductivity of the honeys were determined in accordance with the principles of the harmonised methods of the European Commission for honey [9]. The standard-deviation was calculated for every honey parameter from three replicate assays.

### 2.3. Rheological Assays

The rheological evaluation of the honey in this paper was performed on an MCR 102 rheometer (Anton Paar, Graz, Austria) with a measuring geometry cone-plate. The gap between the cone and the plate was set to a stable value of 0.103 mm. The diameter of the cone was equal to 50 mm with an angle of 1°. The rheological tests were performed at a temperature of 25 °C. Each measurement was repeated four times.

#### 2.3.1. Hysteresis Loop Test

The range of the shear rate was set from 1 to 100 s^−1^ [29]. The hysteresis area was calculated with the use the corporate software RHEOPLUS v. 3.61 (Anton Paar, Austria).

#### 2.3.2. The Determination of Time-Dependent Behaviour

A constant value of 50 s^−1^ of the shear rate was set [30]. The values of the apparent viscosity were recorded for 300 s. Subsequently, the proportions of the apparent viscosity in the 300th and in the 1st second of the test were carried out. The time dependence of the shear stress versus time at a constant shear rate (50 s^−1^) was carried out. The dependence was described by the Weltman model [31]:(1)τ=A−B(lnt)
where:τ—shear stress (Pa)A—value of the shear stress in the first second of the measurement (Pa)B—time coefficient of the thixotropic breakdown (-)t—time of shearing (s)

#### 2.3.3. Dynamic Frequency Sweep Test

The dynamic frequency sweep test was carried out in a frequency range of 0.1–10 Hz at a temperature of 25 °C and at a 1% strain. It was determined by an amplitude sweep test. The amplitude sweep test was determined at 1 Hz in a strain range of 0.1–100%.

#### 2.3.4. Mathematical Modelling

The dependence of the shear stress on the shear rate was described by two non-Newtonian mathematical models. The mathematical models were determined with use MATLAB R2018b (MathWorks, Natick, MA, USA). The first model was the power-law fluid (also known as the Ostwald-de Waele model). The Ostwald-de Waele model is given by following equation [32]:(2)$$\tau =K{\dot{\gamma }}^{n}$$

The second mathematical model was the Herschel-Bulkley rheological model [33]:(3)$$\tau =\tau_{0}+K{\dot{\gamma }}^{n}$$
where:τ—shear stress (Pa)$\tau_{0}$—yield stress (Pa)K—consistency factor (Pa·s^n^)$\dot{\gamma }$—shear rate (s^−1^)n—flow index (-)

### 2.4. Statistical Analysis

The open source software R (v. 3.5.1) [34] was used for the statistical analysis. All the tests were performed at a significance level of α = 0.05. The Shapiro–Wilk test was used to test the hypothesis whether the data are from a normal distribution. A one-factor ANOVA (analysis of variance) was used to test the hypothesis whether there are significant differences between the individual characteristics of a data set. Tukey’s test was used for the post hoc analysis. However, when the hypothesis *h*_0_ about the homogeneity of variance was rejected, Nemenyi’s test was used. The hypothesis *h*_0_ about the homogeneity of variance was tested with use of Leven’s test.

## 3. Results

### 3.1. Hysteresis Area

In the first phase, the dependence of the shear stress on the shear rate was measured. Subsequently, the hysteresis areas were calculated. The course of curve showing the dependence of the shear stress on the shear rate for sample S80 and S100 is obvious from Figure 1. Other samples (S0, S10, S40 and S60) are given in Figure 2. The value of the shear stress at 100 s^−1^ is higher in sample S80 than the shear stress value of sample S100.

The exploratory data analysis of the hysteresis area values for the individual samples is shown in Table 2. The average values of the hysteresis loop are also shown in the Figure 3 for clarity. The sample S0 showed a hysteresis area value despite that it is demonstrably a pure lime nectar honey. The values of a hysteresis area are negative for the samples S0, S10. The value of a hysteresis area grows non-linearly as a proportion of the heather honey increases in the sample. A sudden increase of a hysteresis area value can be seen between samples S20 and S40 (see Figure 3). The values of hysteresis areas showed a high degree of variability, as can be seen from the Table 2. The sample S20 showed the highest value of a variation coefficient. Contrarily, the sample S100 showed the lowest value of a variation coefficient.

The datasets containing hysteresis area values of all the samples were compared with each other by use of an ANOVA analysis and Tukey’s test for the post hoc analysis. The results are shown in Appendix A (Table A1). The hypothesis that the differences between sample S60 and all other samples, sample S80 and all other samples, and S100 and all other samples are not significant was rejected.

In the next step, the mathematical description of the shear stress and shear rate dependence was carried out. For this purpose, the Ostwald-de Waele (OW) and Herschel-Bulkley (HB) models were used. The results of the mathematical modelling are given in Table 3. The flow index n, consistency factor K, possible yield stress *τ*_0_ were determined. In contrast to the hysteresis area values, the other parameters (n, K and $\tau_{0}$) changed adequately with the change in the honey dilution.

The flow index n determines the degree of curvature. This curvature (including the change in the curvature) can also be seen in Figure 1 and Figure 2. The factor n of sample S0 is very close to one in the case of both models. This means that the function describing the dependence of the shear stress on the shear rate is linear and this is a property of Newtonian fluids. As the amount of heather honey increases, the value of n gradually decreases. The values of n<1 are typical for shear-thinning materials. In the case of both models, the decrease in the value of the change in index n is, at first, gradual. Between samples S40 and S60, the change increases significantly (see Figure 4). The index of change in index n between samples S40 and S60 for the OW model is the lowest (i= 0.86), i.e., the change from the previous value is the highest in the data file. In the case of the HB model, the index of change between samples S40 and S60 is i= 0.86 and it is also the highest change in the data file. It can also be observed from Table 3 that the standard deviation of a parameter n is relatively small (despite the very high variability of the hysteresis area values). The maximum value of the coefficient of variation was 2.8% for sample S100.

The change in the consistency factor (K) is also nonlinear. The largest change occurs, as with index n, between samples S40 and S60. For the OW model, the index of change is i= 2.52, for the HB model, it is i= 2.26.

In the HB model, negative yield stress values $\left(\tau_{0}\right)$ were calculated in some cases. For samples S0–S40, the values of $\tau_{0}$ are around the value of 0 Pa. Again, there is a step change between samples S40 and S60.

### 3.2. Dependence of the Apparent Viscosity versus Time

The results of the measurement showing the dependence of the apparent viscosity versus time are graphically presented in Figure 5. In the case of samples S80 and S100, it can be observed that in the first seconds of the measurements, high apparent viscosity values are measured. Subsequently, a sharp decline occurs. Such behaviour is attributed to shear-thinning fluids with thixotropic behaviour. With that, the values of the apparent viscosity in the case of sample S80 are generally higher than the apparent viscosity values of sample S100. It is similar to the case of the values of the hysteresis area. However, if the proportion of the apparent viscosity value in the 300th second of the measurement and the apparent viscosity value in the 1st second of the measurement is set, the situation is changes. This proportion can be marked by the Greek letter ϕ. The following equation for the calculation of ϕ can be introduced:(4)$$\phi =\frac{\eta_{A}^{1}}{\eta_{A}^{300}}$$
where:
$\eta_{A}^{1}$—apparent viscosity in the 1st second of the assay (Pa·s)$\eta_{A}^{300}$—apparent viscosity in the 300th second of the assay (Pa·s)

The results of these data are evident from Table 4. As in the case of parameter n, the change of parameter ϕ is, at first, gradual, then the largest change occurs between samples S40 and S60 (see Figure 6). The change is 13% here (the second highest change is 9%). The Low data variability is also evident, where the value of the variation coefficient ranges from 0.4 to 5.8%.

The hypothesis whether there are significant differences between ϕ was tested. The results are given in Table A2 in Appendix A. A significant difference was found between the ϕ value of sample S100 and all the other samples. The exception is sample S80, where the hypothesis *h*_0_ was not rejected.

The determination of the thixotropic index, using the Weltman model, is necessary for the determination of thixotropy degree. Results are given in Table 5. Coefficient B (time index of thixotropy) is the most important parameter of the Weltman model. The negative value means that the curve grows over time, a positive value means the opposite. It is possible to observe the trend in the results of coefficient B. The significant change again occurs between samples S40 and S60 (see Figure 7). The highest coefficient B value can be seen in the case of sample S100. However, it should be noted that a high degree of variability is possible in the data sets. The greatest variability of data was found in the sample S40, where SD = 2.2 and the coefficient of variation was 96% (see Table 5). It is also clear from the results that the highest values of the coefficients of determination are achieved in sample models S0 and S100, i.e., the original unmixed honey solutions. A significant difference was calculated (ANOVA, Tukey HSD) between sample S100 and samples S0, S10, S20, S40 (see Appendix A, Table A3).

### 3.3. Frequency Sweep Test

The results of frequency sweep test are evident from Figure 8 and Figure 9. The first figure represents the dependence of the loss modulus ($G^{\prime }$) and storage modulus ($G^{\prime \prime }$) on the angular frequency in samples S100 and S60. The second figure represents the dependence of the complex viscosity ($\eta^{*}$) on the angular frequency of S0, S40, S60, S80 and S100.

The results of the frequency sweep test show that the curve storage modulus is gradually approaching the curve of the loss modulus as the proportion of heather honey in the sample increases. In other words, the curves are closest in the case of sample S100. On the contrary, the largest gap can be seen in the case of S0 sample. In the all the cases, the viscous module is dominant.

It is possible to describe the dependence of the complex viscosity ($\eta^{*}$) on the angular frequency by a logarithmic function. This function can be represented by the following equation:(5)$$\eta^{*}=C\left(\ln\omega \right)+D$$
where:ω—the angular frequency (rad·s^−1^)C—the degree of a declining complex viscosityD—the initial value of the complex viscosity at 1 rad·s^−1^

It can be assumed that the higher the slope of the curve, the better the system is structured. Additionally, otherwise, the higher the negative value of the C parameter, the system is better structured. The change of parameter C depending on the sample is evident from the Figure 10. Taking the variability of the data into account, the highest negative value of parameter C is seen in the case of samples S80 and S100 (see Table 6). Significant differences were recorded between samples S100 and the other samples (ANOVA, HSD Tukey), except for samples S80 and S100, where the hypothesis *h*_0_ was not rejected.

## 4. Discussion

A relatively frequently used method for determining the degree of thixotropy is to determine the hysteresis area [29]. It is assumed that the value of the hysteresis area will be zero in the case of a Newtonian liquid (lime nectar honey, sample S0), or it will be close to zero. Conversely, the hysteresis area value, in the case of non-Newtonian fluid (heather honey, sample S100), will take on certain positive values (the samples show structural breakdown), as is clear from other professional work dealing with the rheological properties of heather honey e.g., [1]. It is further assumed that in the case of diluting heather honey by lime nectar honey, the hysteresis area values will gradually decrease as the proportion of lime nectar honey increases.

From the resulting values of the hysteresis area, it is clear that the previously mentioned hypothesis is not quite in accordance with the measured data. The point is, that the mean values of the hysteresis area in the case of sample S0 takes values other than zero, even became negative.

The values of a hysteresis area also often showed the very high degree of a variability at various samples (the highest variation coefficient was approx. 400%, see sample S20 in Table 2). It is, therefore, difficult to accept hysteresis area as an evaluation parameter. These findings need to be commented on in more detail.

The hysteresis loop test is still relatively popular due to its time-saving potential and simplicity. At the same time, however, this test has been criticised from several points of view. For example, the measurements are affected by the inertia effect (although this effect is already often removed using the software that is part of the measuring device). Furthermore, for example, the shear rate is dependent on both shear rate and on time for time-dependent materials. However, in the hysteresis loop test, we are not able to separate the effects of these two variables [35]. The size of the hysteresis area is also dependent on the conditions under which the test is performed, e.g., the shear history prior to the start of the experiment, the maximum shear rate and the acceleration rate [36]. However, it follows that the hysteresis area is a relative parameter dependent on factors that we cannot influence and, most importantly, we are not able to recognise. This is especially true of the sample history. We can never be sure how the material was handled before it was in the laboratory. The time required to restore the structure of the thixotropic material can be very different [37,38]. It should also be noted that honey is a viscous liquid in which solid particles (e.g., pollen grains) are dispersed. These particles settle naturally. The dynamics of these changes may be different for different honeys due to the different densities of the solutions. The dynamics of these processes are also continuously disrupted by taking a new sample when dosing honey into the rheometer. These factors can explain the high variability in the measured values of the hysteresis area. By increasing the number of repetitions, it is possible to reduce the degree of a variability. The number of repetitions may generally seem a weakness in these types of experiments. An above-standard number of repetitions was chosen in the work, i.e., four repetitions. In most works of this type, three repetitions are performed. However, even such a number of repetitions seems to be insufficient. However, the situation cannot be solved by increasing the number of repetitions, as this would be very time consuming and the test sets would not be completed in one day. As mentioned above, this type of sample changes over time, and, therefore, it is possible that over the next few days we would be working with a “different” sample, which would already have different properties than in the previous days, even if only slightly. It also does not seem practical to increase the number of repetitions when there are other (more stable) rheological parameters, which have a potential for the evaluation of the diluted honeys.

Another fact that needs to be clarified is the measured negative value of the hysteresis area in the case of sample S0 (pure lime nectar honey). This type of honey has not been reported as acting as a time dependent substance [8]. However, for example the phenomenon of a thixotropy can occur when honey contains crystals [39,40]. However, the honey samples were thermally thoroughly de-crystallised. The content of the crystals was examined under a light microscope at high magnification (600 times) and with a phase-contrast. Moreover, if there were any crystals in the original sample, they would be present in the all-tested samples and the impact of the crystals on the rheology would be similar in the all samples. In addition, the negative value of a hysteresis area shows rather to shear-thickening fluid with rheopectic behaviour. A closer view of the index n, however, we find that the curve expressing the dependence of the shear stress on the shear rate pointing up (shear rate from 0 s^−1^ to 100 s^−1^) is described by the power function with factor $n_{up}=0.9924\;\pm \;0.007$ (see Ostwald-de Waele model), in the case sample S0. However, the curve pointing down (shear rate from 100 s^−1^ to 0 s^−1^) is described by the power function with factor $n_{down}=0.9692\;\pm \;0.014$. So first the curve is linear, then it arches. This is not a typical property of the shear-thickening fluid with rheopectic behaviour. Taking into account this fact, other measured parameters, and a relatively high degree of variability of the hysteresis loop in the sample S0, the sample S0 cannot be considered as shear-thickening fluid with rheopectic behaviour, but Newtonian fluid.

For the above reasons and also on the basis of information from scientific publications e.g., [36], it is necessary to approach the use of the hysteresis area as a relative parameter for evaluating the degree of the thixotropy of a honey in a somewhat reserved manner, and we are inclined rather not to use HA as a relative evaluation parameter. However, this contradicts the results of the work of [11] who, in turn, state that the area of a hysteresis loop is a suitable parameter for heather honey authentication. This assumption was probably made on the basis of the principal component analysis, where the correlations between the four rheological parameters were determined, with the strongest correlation being between hysteresis area and B (index of the Weltman model). The other parameters in the work [11] are not included in the principal component analysis. Thus, the question is: if the authors [11] had included the other parameters in the analysis, would strong correlations be found between them or not?

In the next step, the parameters of the rheological models were compared. Two rheological models were used: The Herschel–Bulkley model and the Ostwald–de Waele model. These two models are also used in other professional work on rheological properties, the Herschel–Bulkley model was used in [1,40,41] and the Ostwald–de Waele model was used in [11,41]. The results of these studies are summarised in Table 7. If we compare the measured values in sample S100 with the pure heather honey with the values derived from the literature sources, it is clear that the n-values from the literature are higher. The n-value is a relative parameter and the value of this parameter will be influenced by many other factors, as in the case of the hysteresis area. The design of the experiment is an important factor. Some of these are, for example, the maximum achieved value of the shear rate $\left({\dot{\gamma }}_{max}\right)$, the temperature of the sample, etc. The same value $\left({\dot{\gamma }}_{max}\right)$ was set, as in our case, in the works [1,11].

Another parameter that can be compared is the yield stress ($\tau_{0}$). If the yield stress value ($\tau_{0}$), is measured in the sample, a non-Newtonian behaviour can be expected. However, [40] state that the yield stress ($\tau_{0}$) can be even in the case of honeys, which are considered Newtonian. The yield stress is represented as a finite stress required to achieve flow and, at lower temperatures (10, 20 °C), the yield stress occurs as an effect of the microparticles (crystals) in the honey [40]. From the measured values, it is clear that in the case of the gradual addition of heather honey to lime nectar honey, the values of $\tau_{0}$ are first close to zero in the samples with a proportion of heather honey up to 40%. A higher proportion of heather honey led to a sharp increase in the parameter $\tau_{0}$. Negative yield stress values ($\tau_{0}$) are physically unrealistic [42]. However, the yield stress values are estimated based on a function describing the flow curve. For this reason, the yield stress should be interpreted as a model parameter rather than the actual yield stress value, which is a physical property of the fluid itself [1]. In the work [41] negative yield stress values ($\tau_{0}$) were even measured for the pure heather honey.

The parameters n, $\tau_{0}$ showed the non-linear changes in dependence on the dilution as the essential attribute with a significant change between the 40% and 60% dilution. The same applies to other parameters, such as parameter B in the Weltman model, parameter ϕ, or parameter C in the case of the logarithmic dependence of the complex viscosity on the angular frequency.

The reliable parameters for the description of the rheological features of the diluted samples seems to be the n-value according to both models, Ostwald–de Waele and Herschel–Bulkley, and parameter ϕ, in which the most stable parameter with the lowest variability for the characterisation of the thixotropic rate of a honey was proposed.

The standard for heather honey is defined very widely, namely, as far as melissopalynological characteristics is concerned [8]. This situation makes it difficult exactly to decide on the botanical origin of a honey or its violation in spite there are also other chemical-physical markers for heather honey (e.g., specific rheological properties or high content of protein). For instance, a heather honey with a high content of pollen grains (e.g., over 10,000 PG/g and 70% share of the heather PG) can be diluted 1:1 or more while still remaining within the range for the unifloral characteristic for heather honey proposed by [8]. The rheological parameters are relative values and their share in the detection of honey botanic purity has still the characteristic of a supporting, not an arbitrary, method. In the future research, it is appropriate to pay attention to this area, where there is a step change in the measured parameters. It is necessary to examine whether this area changes depending on the choice of the diluent (e.g., different types of honey, special sugar solutions, etc.), and possibly determine how much it changes. If this area were found to be within the same range for different diluents and heather honeys from different localities, this finding could be used to detect honey defects in relation to the purity of the botanical origin. For example, if considered that a step change will be demonstrable in the specified dilution area and testing a random sample of heather honey would show that this area will be shifted towards a lower degree of dilution (i.e., a higher proportion of the random sample than the diluent that may be, e.g., lime nectar honey), this may be a signal that the heather honey has been devalued.

## 5. Conclusions

The area value of the hysteresis loop is sensitive to both external and internal factors. This is an unstable parameter and we do not recommend using it as a comparison parameter.Due to the low variability of the measured values, the relative comparison *n*-value parameter and parameter ϕ appear to be suitable.The dependence of the measured parameters *n*, B, ϕ and C on the degree of the dilution is non-linear and a distinct step change occurs between samples S40 and S60, i.e., the samples that contained 40% (*w*/*w*) heather honey and 60% (*w*/*w*), respectively.

The fact that the dependence of the measured parameters on the degree of the dilution is non-linear with a step change in a certain phase of the dilution is a new finding in the field of the rheological behaviour of honeys. This knowledge could be used, for example, to identify adulterated heather honeys. For this reason, more attention should be paid to the dilution interval between 40% (*w*/*w*) and 60% (*w*/*w*) of heather honey.

It is also important to consider the number of repetitions in this regard. Usually, three replicates are performed in similar experiments. Four replicates were performed in our experiment. However, the mean values calculated from the measurement results from a small number of replicates are prone to higher uncertainty. We can refine the decision-making process by increasing the number of repetitions, but this can lead to difficulties in organising and securing the experiment. The experiment may become unfeasible. Already, the total number of four repetitions was relatively time consuming and, in some cases, this number of repetitions proved to be insufficient (see, for example, the determination of the area of the hysteresis loop). A suitable strategy may be to perform pre-tests with a lower number of repetitions, but on a larger range. Then focus on a narrow area that not behaves as expected and perform an increased number of repetitions in the experiment. In our case, this may be the area defined by the dilution interval of 40% (*w*/*w*) and 60% (*w*/*w*) of heather honey.

## Figures and Tables

**Figure 1 materials-14-02472-f001:**
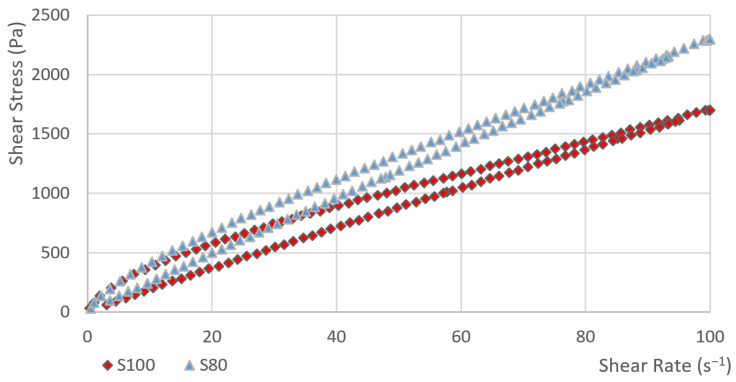
The dependence of the shear stress on the shear rate (sample: S80 and S100).

**Figure 2 materials-14-02472-f002:**
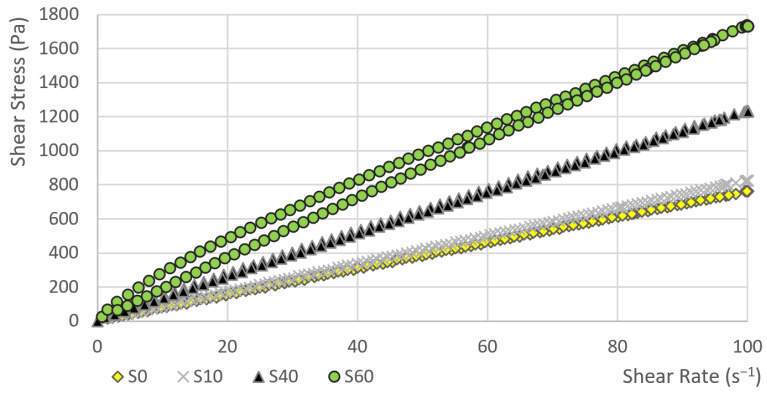
The dependence of the shear stress on the shear rate (sample: S0, S10, S40, S60).

**Figure 3 materials-14-02472-f003:**
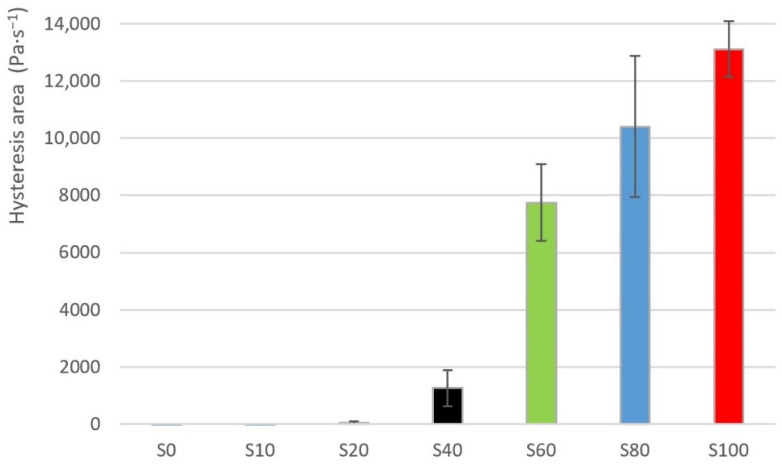
Hysteresis area average values for samples S0–S100.

**Figure 4 materials-14-02472-f004:**
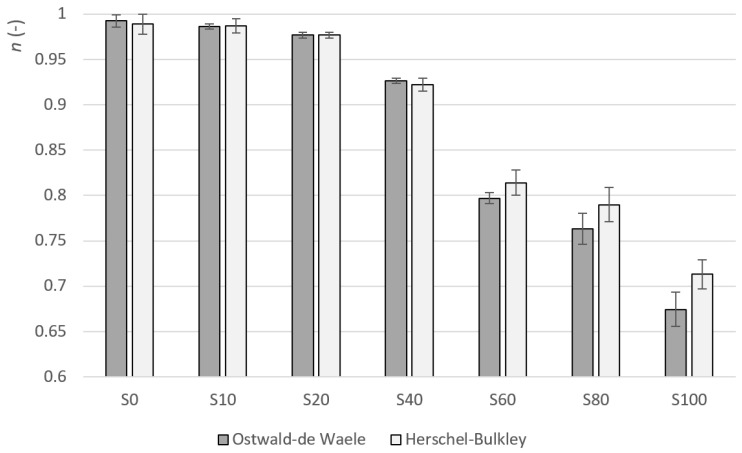
Flow index n average values for samples S0–S100.

**Figure 5 materials-14-02472-f005:**
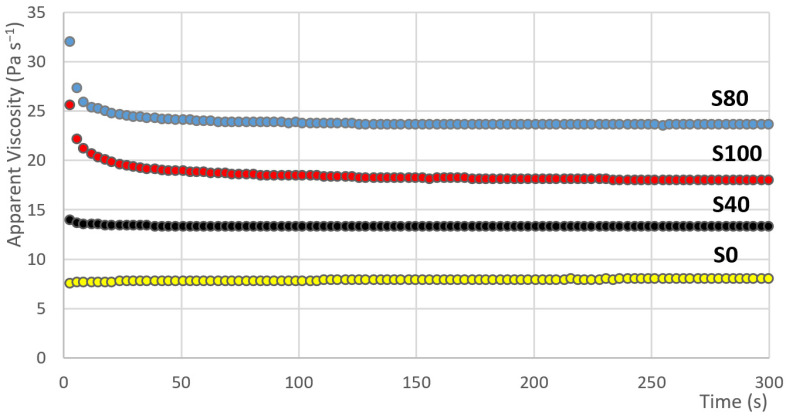
The dependence of the apparent viscosity on the time (samples: S0, S10, S40, S60).

**Figure 6 materials-14-02472-f006:**
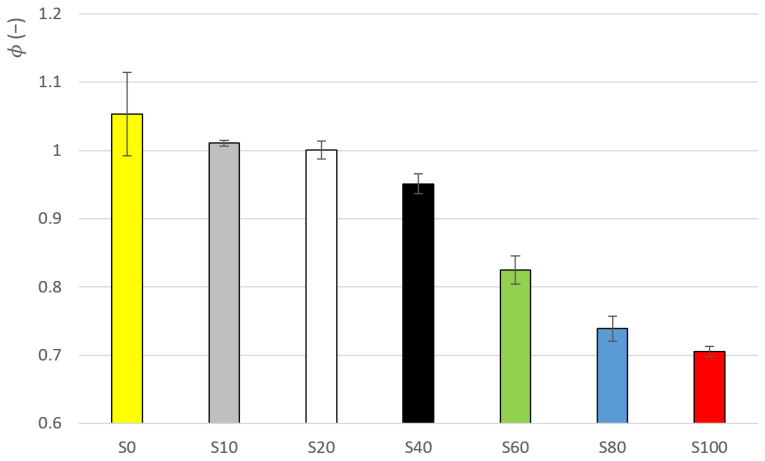
Parameter *ϕ* average values for samples S0–S100.

**Figure 7 materials-14-02472-f007:**
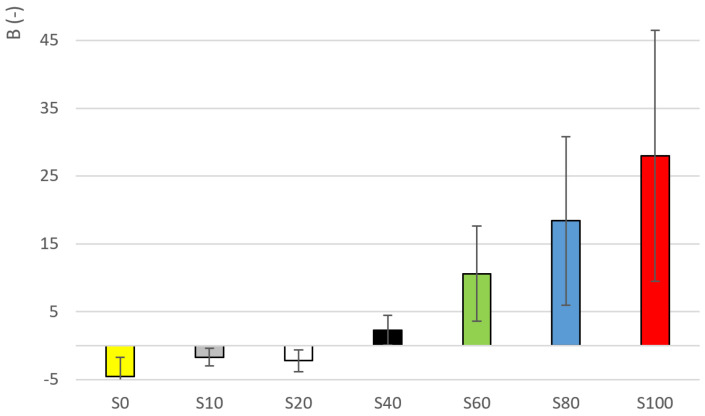
Parameter B values for samples S0–S100.

**Figure 8 materials-14-02472-f008:**
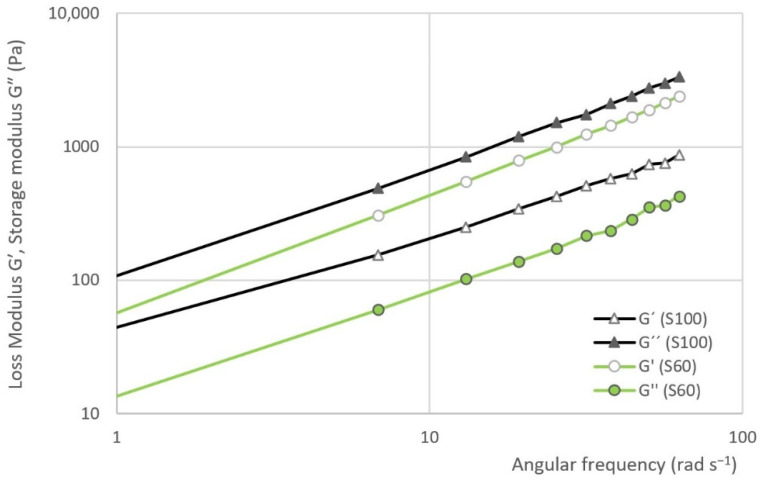
The dependence of the loss modulus ($G^{\prime }$) and storage modulus ($G^{\prime \prime }$) on the angular frequency in the selected samples.

**Figure 9 materials-14-02472-f009:**
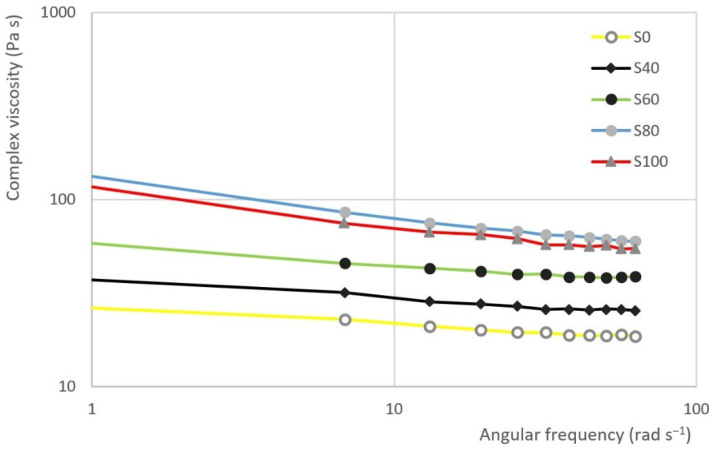
The dependence of the complex viscosity ($\eta^{*}$) on the angular frequency in the selected samples.

**Figure 10 materials-14-02472-f010:**
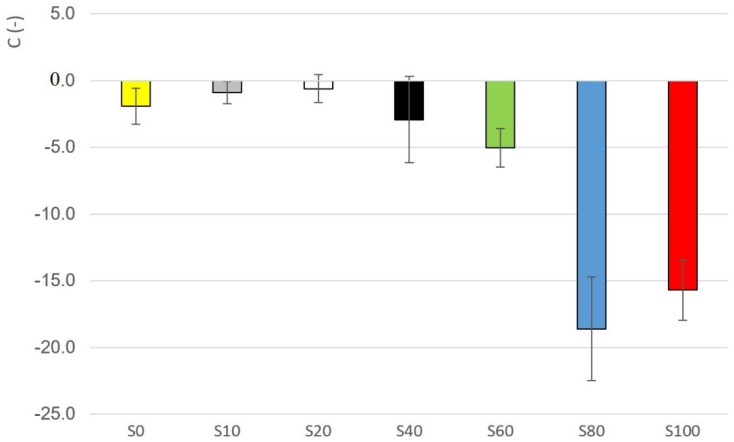
Parameter C values for samples S0–S100.

**Table 1 materials-14-02472-t001:** Description of the honey samples.

Sample	Botanical Origin	Geographical Origin	Pollen Grains in 1 g of Honey	Percentage of Important Pollen Grains	Important Pollen Grains in 1 g of Honey	Honeydew Elements	Water Content (%)	Electrolytic Conductivity (mS·cm^−1^)
PA/386	heather nectar honey	Norway	8886 ± 467	28.1 ± 1.2	1786 ± 21	sporadic clusters, no bodies	16.9 ± 0.2	86.6 ± 4.2
PA/416	lime nectar honey	Czech Republic	2252 ± 139	9.6 ± 0.5	216 ± 1	sporadic clusters, no bodies	17.8 ± 0.1	50.4 ± 1.9

**Table 2 materials-14-02472-t002:** Exploratory data analysis—hysteresis area.

Sample	Mean (Pa·s^−1^)	Median (Pa·s^−1^)	SD (Pa·s^−1^)	VC (%)	IQR (Pa·s^−1^)	S-W (-)
S0	−493.87	−532.96	240.619	−48.7208	320.631	0.72
S10	−217.28	−245.97	69.333	−31.9095	82.403	0.05
S20	20.39	0.67	81.724	400.7579	112.616	0.54
S40	1254.58	1476.30	629.889	50.2071	769.839	0.14
S60	7742.53	7386.65	1342.529	17.3397	2020.350	0.39
S80	10,409.83	10,577.50	2469.510	23.7229	3373.350	0.91
S100	13,124.50	12,859.00	974.223	7.4229	1473.000	0.35

SD—standard deviation, VC—variation coefficient, IQR—interquartile range, S-W—Shapiro-Wilk test (the measurement of each sample was performed in four replicates).

**Table 3 materials-14-02472-t003:** Resulting parameters of the rheological models.

Ostwald-De Waele Model				Herschel-Bulkley Model			
Sample	K (Pa·s^n^)	*n* (-)	R^2^ (-)	$\tau_{0}$ (Pa)	K (Pa·s^n^)	*n* (-)	R^2^
S0	7.91 ± 0.3	0.9924 ± 0.007	1.0000	−1.15 ± 2.2	8.0 ± 0.4	0.9891 ± 0.011	1.0000
S10	8.79 ± 0.3	0.9864 ± 0.003	1.0000	0.29 ± 1.7	8.8 ± 0.5	0.9872 ± 0.008	1.0000
S20	10.35 ± 0.1	0.9768 ± 0.003	1.0000	−0.02 ± 1.2	10.4 ± 0.2	0.9768 ± 0.003	1.0000
S40	17.39 ± 1.1	0.9265 ± 0.003	1.0000	−2.86 ± 2.8	17.8 ± 1.4	0.9220 ± 0.007	1.0000
S60	43.80 ± 2.1	0.7970 ± 0.006	0.9998	17.28 ± 7.8	40.2 ± 2.3	0.8141 ± 0.014	0.9998
S80	67.22 ± 6.1	0.7634 ± 0.017	0.9995	36.45 ± 6.7	58.8 ± 5.5	0.7898 ± 0.019	0.9997
S100	74.50 ± 6.2	0.6745 ± 0.019	0.9986	44.94 ± 22.9	61.0 ± 6.4	0.7134 ± 0.016	0.9990

±standard deviation (the measurement of each sample was performed in four replicates).

**Table 4 materials-14-02472-t004:** Exploratory data analysis of the ϕ values.

Sample	Mean (-)	Median (-)	SD (-)	VC (%)	IQR (-)	S-W (-)
S0	1.0534	1.0260	0.0610	0.0338	5.7936	0.01092
S10	1.0108	1.0110	0.0043	0.0062	0.4218	0.48
S20	1.0005	1.0049	0.0131	0.0143	1.3060	0.1875
S40	0.9514	0.9522	0.0143	0.0233	1.4984	0.1185
S60	0.8251	0.8258	0.0209	0.0302	2.5322	0.5364
S80	0.7390	0.7390	0.0184	0.0297	2.4838	0.1605
S100	0.7054	0.7074	0.0075	0.0075	1.0650	0.4843

SD—standard deviation, VC—variation coefficient, IQR—interquartile range, S-W—Shapiro-Wilk test (the measurement of each sample was performed in four replicates).

**Table 5 materials-14-02472-t005:** Parameters of the Weltman model.

Sample	A (Pa)	B (-)	R^2^
S0	368.5 ± 14.2	−4.5 ± 2.8	0.83 ± 0.13
S10	441.3 ± 9.4	−1.7 ± 1.3	0.68 ± 0.17
S20	515.9 ± 27.0	−2.2 ± 1.6	0.55 ± 0.34
S40	673.7 ± 22.7	2.3 ± 2.2	0.76 ± 0.19
S60	864.5 ± 111.4	10.6 ± 7.0	0.75 ± 0.17
S80	1284.8 ± 72.2	18.4 ± 12.4	0.73 ± 0.21
S100	1064.2 ± 75.6	28.0 ± 18.5	0.84 ± 0.19

±standard deviation (the measurement of each sample was performed in four replicates).

**Table 6 materials-14-02472-t006:** The parameters of the model describing the logarithmic dependence of the complex viscosity $\left(\eta^{*}\right)$ on the angular frequency (ω).

Sample	C (-)	R^2^
S0	−1.93 ± 1.34	0.81 ± 0.2
S10	−0.91 ± 0.85	0.66 ± 0.5
S20	−0.61 ± 1.05	0.31 ± 0.4
S40	−2.93 ± 3.22	0.92 ± 0.1
S60	−5.05 ± 1.43	0.95 ± 0.01
S80	−18.6 ± 3.90	0.95 ± 0.002
S100	−15.7 ± 2.25	0.95 ± 0.01

±standard deviation (the measurement of each sample was performed in four replicates).

**Table 7 materials-14-02472-t007:** Heather honeys’ parameters by various authors.

Herschel-Bulkley				Ostwald-de Waele					
Source	*n* (-)	K (Pa·s^n^)	$\tau_{0}$ (Pa)	*n* (-)	K (Pa·s^n^)	t (°C)	ϕ (%)	HA (Pa·s^−1^)	${\dot{\gamma }}_{max}$ $\;(s{-}^{1})$
(A)	0.70 ± 0.01	50.7 ± 1.2	50.2 ± 1.3	-	-	20	18.0	15,000	100
	0.77 ± 0.01	29.1 ± 0.3	50.2 ± 1.9	-	-	20	18.2	7000	
	0.88 ± 0.01	10.8 ± 0.6	3.8 ± 0.2	-	-	20	20	2000	
(B)	0.901	13.39	0.15	-	-	20	18.7	-	50
(C)	0.988	112	−0.64	0.996	4.71	30	24.0	-	5
(D)	-	-	-	0.88 ± 0.1	23.67 ± 8.0	25	17.5	6994 ± 1945	100
(A)	0.70 ± 0.01	50.7 ± 1.2	50.2 ± 1.3	-	-	20	18.0	15,000	100

t—temperature, φ—humidity, HA—hysteresis area, ${\dot{\gamma }}_{max}$—maximal value of shear rate. (A)—Ref. [1], (B)—Ref. [40], (C)—Ref. [41], (D)—Ref. [11].

## Data Availability

The data presented in this study are available on request from the corresponding author.

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
