# Peer review of "A Rheological Analysis of Biomaterial Behaviour as a Tool to Detect the Dilution of Heather Honey"

_materials, 2021, doi:10.3390/ma14102472_

Round 1

Reviewer 1 Report

Submitted manuscript "A rheological analysis of biomaterial behaviour as a tool to detect the dilution of heather honey" deals with analysis of heather honey with mixture of common honey.  

Overall, ms is interesting and well written. However, there are some concern about the methods.

Why did authors choose lime (linden) honey, that is also valuable type of honey and did not chose other type with low commercial value or alternatively used syrup to mix with heather honey.

Each method mentioned in the ms should have appropriate reference (source).

Furthermore, only one sample of heather honey was used and this represents a weakness of the study. It would be great to have more samples of heather honey with different geographical origin. 

Author Response

The detailed response to the reviewer's comments is attached.

Reviewer 2 Report

In my previous review of the article I  stressed that:

‘’Dear Editors,

Manuscript materials-1204126 presents a novel physicochemical analysis applied to heather honey, concerning   a rheological analysis complemented with mathematical models, as a tool to detect the dilution of heather honey. Data obtained are important to be used for the adulteration control of honey. Thee article is well written, and results are clearly given. I have, thus, some minor suggestions for authors.

-Line 443   and elsewhere. ‘’…by Persano Oddo et al. [8]’’. 

-Line 449. Kindly check the English language.

-Line 469 and elsewhere. Change ‘’disrupted’’ to adulterated.

Based on the above, I suggest a minor revision prior to further consideration for publication.

I also pointed that ‘’The research is designed appropriately. I don't have any other comments concerning other sections of the paper. As I have already mentioned in my original review the results are clearly presented’’.

During my new review, I show that the authors did not cover my comments. In fact, these were some comments that could be easily addressed. Given the time a Reviewer spends to evaluate an article, the suggestions should be covered.

Based on the  above, I leave the final decision to the Editors.

Author Response

(The authors gave the same response as above.)

Reviewer 3 Report

The manuscript by Přidal et al investigated the application of using rheological analysis as a tool to determine the dilution of heather honey with honey from other botanical origin. Several rheological parameters were measured and compared to determine which set of parameters can be best suited as comparative parameters. Below are my major comments:

1).  I would suggest replacing  most of the tables with figures (bar graph would work well here), or add figures. The reason is that it is easier for the readers to see the difference between different groups with SD bars and statistical analysis labeled in a figure as compared with Tables with all numbers. Figures will also help a lot to illustrate whether the trends of rheological parameters change with increasing content is linear or non-linear, and whether these change is statistically significant. Also, the non-linear with a step change between 40% and 60%, as the authors mentioned, will likely be demonstrated.

2). From Table 1, it seems to me (little knowledge and experience with honey industry) that the lime and heather honey are very different from each other in terms of pollen grains etc. In this case, can I assume that it might make it technically easier to differentiate pure heather honey with heather honey diluted with lime honey? I understand that it will take more time and efforts, but it will be convincing if the authors can demonstrate that these set of parameters will also work for heather honey diluted with a 2ndand 3rd honey type.

3). When samples (S0, S10. etc.) were first mentioned in line 172, it is not clear to the readers what are these refer to. Will have to look for descriptions in the text below to understand, e.g. whether S0 or S100 is 100% heather honey.

Author Response

(The authors gave the same response as above.)
